# Sensitivity of the *CHIMERE* model to changes in model resolution and chemistry over the northwestern Iberian Peninsula

Swen Brands<sup>1</sup>, Guillermo Fernández-García<sup>1</sup>, Marcos Tesouro Montecelo<sup>1</sup>, Nuria Gallego Fernández<sup>2</sup>, Anthony David Saunders Estévez<sup>2</sup>, Pablo Enrique Carracedo García<sup>1</sup>, Anabela Neto Venancio<sup>1</sup>, Pedro Melo Da Costa<sup>1</sup>, Paula Costa Tomé<sup>2</sup>, Christina Otero<sup>2</sup>, María Luz Macho<sup>1</sup>, and Juan Taboada<sup>1</sup> <sup>1</sup>MeteoGalicia - Consellería de Medio Ambiente, Territorio e Vivenda, Xunta de Galicia, Santiago de Compostela, Spain <sup>2</sup>Servicio de Calidad del Aire - Consellería de Medio Ambiente, Territorio e Vivenda, Xunta de Galicia, Santiago de Compostela, Spain

Correspondence: Swen Brands (swen.brands@gmail.com)

**Abstract.** Here, the capability of the chemical weather forecasting model CHIMERE (version 2017r4) to reproduce summertime surface ozone, particulate matter and nitrogen dioxide concentrations in complex terrain is investigated. The study area is the northwestern Iberian Peninsula, where both coastal and mountain climates can be found in direct vicinity and a large fraction of the land area is covered by forests. Fed by lateral boundary conditions from the ECMWF Composition Integrated

- Forecast System, meteorological data from the Weather Research and Forecasting Model (WRF) and the HTAP v2.2 emission inventory, CHIMERE's performance compared to observations is tested with a range of sensitivity experiments, exploring the role of horizontal and vertical resolution and the effects of applying distinct chemistry mechanisms. Using a high horizontal *and* vertical resolution yields the most balanced verification results. If both the daily maximum and minimum values are important for the given application, then the full Melchior mechanism should be used. If, however, the daily maxima are considered more
- important than the minima, SAPRC should be used instead. In any case, model performance for nitrogen dioxide is clearly not satisfactory for our study region, probably indicating deficiencies in the emission inventory.

# 1 Introduction

Motivated by the air quality legislation of the European Union (EU, 2008), many governmental air quality departments are currently demanding air quality forecasting schemes based on numerical models (Thunis et al., 2016), and the need for accurate and computationally efficient predictions in this field is perhaps greatest than ever before. For Europe as a whole, the most important real-time prediction system available to date is provided by the Copernicus Atmosphere Monitoring Service (Marécal et al., 2015), comprising an ensemble of currently seven chemical weather forecasting (CWF) models<sup>1</sup> run for the entire continent at a horizontal resolution of  $0.1^{\circ}$ . to  $0.25^{\circ}$  in longitude and  $0.1^{\circ}$  to  $0.2^{\circ}$  in latitude. In addition to this short-term

<sup>&</sup>lt;sup>1</sup>see Kukkonen et al. (2012) for an overview of these models

prediction system, several large research initiatives have been issued during the last two decades in order to assess the *cli-matological* properties of atmospheric composition, including the assessment of long-term tendencies resulting from emission reductions induced by the Convention on Long Range Transboundary Air Pollution (CLRTAP). The final aim of these efforts is to find model configurations, or ensembles thereof, that can be used as surrogates for real observations in order to assess whether emission reductions actually have lead, or would lead, to changes in the atmosphere's composition on climatological

whether emission reductions actually have lead, or would lead, to changes in the atmosphere's composition on climatological time-scales (Vautard et al., 2006; Jonson et al., 2006; Colette et al., 2011; Wilson et al., 2012; Banzhaf et al., 2015; Colette et al., 2017; Im et al., 2018b, a; Vivanco et al., 2018; Theobald et al., 2019).

Complementary to the aforementioned large-scale efforts, usually conducted with a single configuration of a given model, small-scale sensitivity tests for particular models are still relevant since they can be run with more sophisticated model con-

- figurations than their large-scale counterparts and are therefore more interesting for regional prediction systems, such as those demanded by national or regional governments (Banzhaf et al., 2012; Beegum et al., 2016; Flamant et al., 2018). Further, following the concept of seamless prediction (Palmer et al., 2008), lessons learned from short-term prediction systems for relatively small geographical areas might as well be important for longer lead-times and larger areas.
- Previous sensitivity studies have identified several factors influencing the models' capability to correctly reproduce observed values, hereafter referred to as "model performance" (Giorgi and Francisco, 2000; Chang and Hanna, 2004). Among these factors, the meteorological input used to drive these models and the accuracy of the underlying emission inventory play a key role and have been assessed in a number of studies (Menut, 2008; Markakis et al., 2015; Colette et al., 2017; Otero et al., 2018; Vivanco et al., 2018). The resolution of the model mesh used to discretize the chemical reactions and atmospheric dynamics is also important and, when it is increased, a trade-off between potential performance gains and computational cost
- must been made in practice. In what concerns the *horizontal* resolution, performance gains have been reported up to a scale of approximately 12 km for a number of models, such as WRF-CHEM and CHIMERE (Valari and Menut, 2008; Schaap et al., 2015; Crippa et al., 2017). However, a further resolution increase does not guarantee further performance gains. Namely, beyond the 12 km threshold, Misenis and Zhang (2010) reported heterogeneous results for WRF-CHEM that strongly depend on the considered time period. For the use of CHIMERE and focussing on surface O<sub>3</sub> concentrations, Valari and Menut (2008)
- even found a performance loss which they attributed to a noise increase in the emission fluxes and meteorological parameters at higher resolution. Regarding the role of *vertical* resolution, an increase therein has been found to to improve the modelled particulate matter (PM) concentrations during desert dust events when using WRF-CHEM (Teixeira et al., 2016). However, CHIMERE's performance was found to be only weakly affected by such an resolution increase (Menut et al., 2013; Markakis et al., 2015).
- Representing the number and complexity of the considered chemical reactions, several *chemistry mechanisms* are usually available for a given model and switching from one mechanism to another can also affect the model's performance. Model sensitivity to this parameter has been found to be appreciable for WRF-CHEM (Balzarini et al., 2015; Karlický et al., 2017). In recent CHIMERE versions, the SAPRC-07A mechanism (hereafter: SAPRC) has been included as an alternative to the full or reduced versions of the Melchior mechanism (Mailler et al., 2017) but, to the authors' knowledge, related sensitivity tests
- are sparse to date.

A common shortcoming of small-scale sensitivity studies is that the conclusions therein only hold for specific regions, time periods or seasons of the year. In this context, most of the aforementioned conclusions for CHIMERE (the model applied here) have been drawn for the *Île de France* region, which is densely populated, relatively flat and not directly influenced by sea-salt emissions. The model has been applied for a number of other regions, but the map is still incomplete and sensitivity testing is not the main focus of the corresponding studies (Mazzeo et al., 2018; Menut et al., 2018; Monteiro et al., 2018; Brasseur et al.,

2019; Deroubaix et al., 2019).

This is where the present study comes into play: A series of sensitivity test has been run with CHIMERE over the *northwestern Iberian Peninsula*, a region characterized by a complex coastline, forested mountain terrain and the advection of sea-salt from the surrounding Atlantic Ocean. Thus, our study region is quite different from the *Île de France* region. The applied tests

- will isolate the effects of an increase in the model's horizontal and/or vertical resolution, as well the effects arising from a the use of distinct chemical mechanisms (full Melchior or SAPRC). To this end, version 2017r4 of the CHIMERE model is used (Mailler et al., 2017) in combination with the HTAP emission inventory version 2.2 (Janssens-Maenhout et al., 2015). Saharan dust intrusions are *not* accounted for by running CHIMERE on a large domain covering all relevant dust sources (Bessagnet et al., 2017), but by using a far smaller domain ingesting the global forecasts provided by the European Centre for Medium-
- Range Weather Forecasts (ECMWF) Composition Integrated Forecasting system (C-IFS) at its lateral boundaries (Flemming et al., 2015). This strategy largely reduces the computational costs and is an interesting alternative to simulating remote mineral dust emissions, e.g. originating in the Sahara desert, with the CHIMERE model itself (Bessagnet et al., 2017).

In Section 2, the applied data and model configurations used for sensitivity testing are described. Results are presented in Section 3 and a discussion and some general conclusions are provided in Section 4.

# 20 2 Data and Methods

This section opens with a description of the meteorological input data and the general characteristics of the CHIMERE experiments. Then, the particularities of the individual experiments are presented and the in-situ station network used for as reference for verification is introduced. The section closes with a description of the verification measures used to estimate CHIMERE's performance.

# 25 2.1 Meteorological Input and General Characteristics of the CHIMERE Experiments

The meteorological input data for the CHIMERE experiments are provided by the Weather Research and Forecasting (WRF) model version 3.5 (Skamarock et al., 2008), driven by Global Forecast System (GFS) forecasts initialized at 00 UTC (Caplan et al., 1997). WRF is run on three domains, a continental-scale domain having a resolution of 36km, followed by a regional domain covering southwestern Europe at a resolution of 12km and, finally, a 4km domain covering our study region, the

30 northwestern Iberian Peninsula. For these domains, WRF is executed with a minimum time step of 216, 72 and 24 seconds and a maximum time step of 360, 180 and 60 seconds, respectively. All domains comprise 33 vertical layers with a model top at 10 hPa. A detailed overview of the WRF physics can be found in Table 1.

**Table 1.** WRF physics common to all sensitivity tests

| Parameter                | Option                           |
|--------------------------|----------------------------------|
| Microphysics             | WRF single-moment 6-class scheme |
| Longwave radiation       | Rapid Radiative Transfer Model   |
| Shortwave radiation      | Dudhia scheme                    |
| Surface layer            | MM5 similarity                   |
| Land surface             | 5-layer thermal diffusion        |
| Planetary boundary layer | Yonsei University scheme         |
| Cumulus                  | Kain-Fritsch scheme              |

With this meteorological input, version 2017r4 of the CHIMERE model (Mailler et al., 2017) is run in combination with the HTAP v2.2 inventory, representing anthropogenic emissions of the year 2010 (Janssens-Maenhout et al., 2015). Two domains are used for the CHIMERE model integrations: a coarse domain having a horizontal resolution of  $0.15^{\circ} \times 0.15^{\circ}$  (*longitude* × *latitude*), and a fine domain, nested into the former, having a resolution of  $0.05^{\circ} \times 0.04^{\circ}$  (see Figure 1a). Note that the terms

- "coarse" and "fine" shall hereafter refer to the CHIMERE domains, not the WRF domains, if not otherwise stated. Regridding the HTAP dataset to these domains has been accomplished with the emiSURF program without using downscaling with traffic or population proxies (Mailler et al., 2017). Biogenic emissions are from the MEGAN model version 2.04 (Guenther et al., 2006) and mineral dust emissions within the CHIMERE domains are calculated on the basis of the United States Geological Survey landuse dataset (Loveland et al., 2000). The Alfaro and Gomes (2001) saltation and sandblasting scheme, optimized by
- Menut et al. (2005), and the surface wind threshold described in Shao and Lu (2000) are used throughout all experiments. The effect of soil moisture on dust emissions (Fécan et al., 1998) is activated and so are sea-salt emissions. Vertical advection is achieved by the upwind scheme, horizontal advection by the more complex van Leer (1979) scheme. Carbonaceous species and interaction between aerosols and gases are taken into account by the model chemistry. The number of Gauss-Seidel iterations is set to 3, mainly because the model occasionally develops unrealistic waves with lower numbers. Urban correction/reduction
- of the wind speed and the resuspension process are deactivated. A complete list of the internal CHIMERE parameters common to all sensitivity experiments is provided in Table 2. For a full description of these parameters, the interested reader is referred to the CHIMERE user manual available at http://www.lmd.polytechnique.fr/chimere.

Along the lateral boundaries of the coarse domain, the concentrations of the chemical species required by CHIMERE are provided by the forecasts of the ECMWF Composition Integrated Forecasting System (C-IFS) (Flemming et al., 2015), initial-

ized at 00 UTC. This global model comprises 60 vertical levels and has a horizontal resolution of  $\approx 80 km$ . In case a chemical species required by CHIMERE is not provided by C-IFS, the monthly climatological mean values from the MACC reanalysis (Inness et al., 2013) are used instead. Note that the time-varying dust data from C-IFS are scaled by a factor of 0.6 for all experiments since, otherwise, PM concentrations would be overestimated during the two Saharan dust events occurring in the time period considered here (summer 2018).

Table 2. CHIMERE parameters common to all sensitivity test

| Parameter                                         | Option                                       |
|---------------------------------------------------|----------------------------------------------|
| Nr. Gauss-Seidel iterations                       | 3                                            |
| Chemical time-step                                | adaptive                                     |
| Physical time-step                                | 5 minutes                                    |
| Nr. of aerosol size sections                      | 9                                            |
| Chemically-active aerosols                        | yes                                          |
| Anthropogenic emissions                           | HTAP v2.2 for the year 2010                  |
| Sea-salt emission parameterization                | inert, parametrization 0                     |
| Biogenic emissions                                | MEGAN                                        |
| Mineral dust emission                             | On                                           |
| Saltation and sandblasting scheme                 | Alfaro and Gomes (2001), Menut et al. (2005) |
| Wind threshold estimation                         | Shao and Lu (2000)                           |
| Effect of soil moisture on mineral dust emissions | Fécan et al. (1998)                          |
| Secondary organic aerosol scheme                  | medium complexity                            |
| ISORROPIA coupling                                | yes                                          |
| Inclusion of carbonaceous species                 | yes                                          |
| Aerosol dry deposition                            | Zhang et al. (2001)                          |
| Horizontal advection scheme                       | van Leer                                     |
| Vertical advection scheme                         | upwind                                       |
| Urban correction                                  | off                                          |
| Resuspension process                              | off                                          |
| Deep convection                                   | on                                           |
| Land cover dataset                                | USGS                                         |
| Lateral boundary conditions                       | time-varying C-IFS data                      |

5