# Peer review of "Sensitivity of the *CHIMERE* model to changes in model resolution and chemistry over the northwestern Iberian Peninsula"

_Atmospheric Chemistry and Physics, 2019_

## Referee Comment (RC1) · Anonymous Referee #1 · 19 Jun 2019

The manuscript of Brands et al. investigates the sensitivity of the CHIMERE model to different horizontal and vertical resolutions and to different chemical mechanisms. The focus is on a region of the northwestern Iberian Peninsula where the CHIMERE model has not been applied so far. The model results are evaluated against observations with a focus on minimum and maximum values.

To my opinion the current version of this manuscript does not fit into the scope of ACP(D) as it is mainly a technical analysis of the model system and has no broader scope or general implications for atmospheric science (see aims and scope of ACP;

[Figure]

'The journal scope is focused on studies with general implications for atmospheric science rather than investigations that are primarily of local or technical interest'). To fit into the scope of ACP(D) a more general geoscientific conclusion/scope of the manuscript would be necessary. However, I think that the technical analysis of the model system is an interesting topic and therefore I propose to revise the manuscript largely (see detailed suggestions below) and hand the manuscript over to the partner journal Geoscientific Model Development (GMD). If the authors wish to publish their manuscript in ACP(D) the scope of the journal should be taken better into account and the manuscript needs major revisions.

General comments:

In p4l6 the authors write '[...]without using downscaling with traffic or population proxies[...]'. Accordingly, also for the finest nest with $0.05° \, x \, 0.04°$ resolution, the authors applied emissions with a resolution of $0.1° \, x \, 0.1°$, right? If this is the case this is an important limitation of the study and needs to be clearly stated. It is well known that the emission resolution influence the results largely. A too coarse resolution of the emissions can also deteriorate the model results (e.g. Markakis et al., 2015). Therefore, I propose to perform additional model runs with a downscaling of the emission as this is a general feature of the CHIMERE model. These additional runs can then be used to quantify uncertainties due to missing downscaling of emissions.

The authors find a poor performance for NO2 of the CHIMERE model and link this poor performance to deficits of the emission inventory. To my opinion this argument needs a more detailed investigation. Several things should be discussed/considered:

- The emission inventory is for the year 2010. What was the emission changes in the last 8 years? What trend do ground-level measurements of NO2 show?

Are there more up to date emission inventories available (for example from TNO, EDGAR etc.)? If so, why don't use them?

- What is the performance of NO2 at the different stations types? How well does the model perform at the 'background' stations? How well at the 'traffic' and 'industry' stations?

- Where are the 'traffic' stations located? Does it make sense to evaluate a model with resolutions of 4 to 5 km and emissions at around 10 km resolution with measurements at the street scale? I guess it makes more sense to evaluate against the measurements at (urban) background stations or average all values of the 'traffic' stations of one particular city.

- How well does WRF reproduce the observed meteorology? To efficient mixing of the boundary layer might cause problems in reproducing the measurements. Please provide at least a basic meteorological evaluation of the used meteorological data.

The authors focus only on daily minimum and maximum values. I agree that especially the maximum values are very important with respect to air quality issues. However, to my opinion it would be very important to investigate also the general ability of the model to represent the hourly variability of the measurements. Therefore, I propose to further perform statistical analysis of the whole time series for each station and not only for minimum and maximum values. Further, the analysis does not take into account that the model concentrations could be shifted geographically (e.g. minima and maxima are misplaced due to coarse resolution of the emissions). Therefore, I propose to provide additionally overlay plots (maybe only in the supplement) combining the geographical distribution of the modelled concentration and the measured concentrations as an example see Fig. 5 of Knote et al, 2011)

I guess that the model runtime (and needed resources) of the different experiments differ heavily. It would therefore be very beneficial to confront the performance of the model with of the different configurations with the model runtime and to give recommendations about the trade-off between model runtime and model performance. This could have important implications for other people using CHIMERE. Further, if the authors (or others) plan to use the CHIMERE model for operational forecasts this trade-off would be very important information.

The description of the performed sensitivity studies (Sect. 2.2) is much too short. Readers familiar with the CHIMERE system might be able to follow the description of the authors; readers from outside the 'CHIMERE world' are lost. Please provide more details about the two different vertical grids (e.g. by a figure showing the different levels). Further, please describe the differences of the two chemical mechanisms in more detail. How do they differ? I know that Mailler et al., 2017 provides some details, but details which are very important for this study should be repeated in the manuscript. Further, Menut et al., 2013 already provide a short comparison of the MELCHIO2 and SAPRC07 mechanisms. How do the findings from the authors compare to the findings of Menut et al., 2013?

Specific comments:
(The line numbers in the manuscript seem to be wrong, at least on page 8). I here refer to the line numbers given in the manuscript)

Abstract: Please provide information about the period consider for this investigation in the abstract (e.g. 20.7.2018-31.8.2018).

Section 2.1: Studies show that ozone on the Iberian Peninsula is heavily influenced

by long range transport (e.g. Pay et al., 2019). Hence, boundary conditions are very important in this region. Therefore, please provide more information about the temporal update frequency of the boundary conditions in Sect. 2.1. Further, the authors mention that chemical boundary conditions stem from different systems (C-IFS, MACC). Further, the meteorological boundary conditions for WRF stem from GFS. Please provide short discussions about the influence of inconsistent chemical (and meteorological) boundary conditions. Further, the authors mention that dust information from C-IFS needed to be scaled. What about the other components? Please provide short information about the quality of the chemical boundary data for the investigated period.

P6l10 The authors mention that for dust there is only a minor benefit when the model top height is increased. For ozone, however transport from the stratosphere is a very important feature which is missing in the applied set-up. Please comment on this issue.

P8l5 How did the authors sample the model data? The authors took the results at the lowest level in the corresponding grid box, right? But did the authors chose instantaneous model results or temporal (e.g. hourly averaged) model output?

P9l21 Especially in complex terrain the height of the lowest model layer and the height of the station might not fit together. Therefore, please provide a comparison of the station height and the height of the model at the lowest layer and check how large the differences are.

Technical corrections:
Figure2: Please fix the legend ('hores')

**Bibliography**

Knote, C., Brunner, D., Vogel, H., Allan, J., Asmi, A., Aïjalä, M., Carbone, S., van der Gon, H. D., Jimenez, J. L., Kiendler-Scharr, A., Mohr, C., Poulain, L., Prevot, A. S. H., Swietlicki, E., and Vogel, B.: Towards an online-coupled chemistry-climate model: evaluation of trace gases and aerosols in COSMO-ART, Geosci. Model Dev., 4, 1077-1102, https://doi.org/10.5194/gmd-4-1077-2011, 2011.

Mailler, S., Menut, L., Khvorostyanov, D., Valari, M., Couvidat, F., Siour, G., Turquety, S., Briant, R., Tuccella, P., Bessagnet, B., Colette, A., Letinois, L., Markakis, K., and Meleux, F.: CHIMERE-2017: from urban to hemispheric chemistry-transport modeling, Geosci. Model Dev., 10, 2397-2423, https://doi.org/10.5194/gmd-10-2397-2017, 2017.

Markakis, K., Valari, M., Perrussel, O., Sanchez, O., and Honore, C.: Climate-forced air-quality modeling at the urban scale: sensitivity to model resolution, emissions and meteorology, Atmos. Chem. Phys., 15, 7703-7723, https://doi.org/10.5194/acp-15-7703-2015, 2015.

Menut, L., Bessagnet, B., Khvorostyanov, D., Beekmann, M., Blond, N., Colette, A., Coll, I., Curci, G., Foret, G., Hodzic, A., Mailler, S., Meleux, F., Monge, J.-L., Pison, I., Siour, G., Turquety, S., Valari, M., Vautard, R., and Vivanco, M. G.: CHIMERE 2013: a model for regional atmospheric composition modelling, Geosci. Model Dev., 6, 981-1028, https://doi.org/10.5194/gmd-6-981-2013, 2013.

Pay, M. T., Gangoiti, G., Guevara, M., Napelenok, S., Querol, X., Jorba, O., and Perez Garcia-Pando, C.: Ozone source apportionment during peak summer events over southwestern Europe, Atmos. Chem. Phys., 19, 5467-5494, https://doi.org/10.5194/acp-19-5467-2019, 2019.

---

## Referee Comment (RC2) · Anonymous Referee #2 · 15 Jul 2019

Quite a few different models are currently in use for chemistry transport modelling on the regional scale. Still many questions concerning the validity of the model results with respect to the necessary complexity of the chemistry mechanisms, the needed quality of underlying emissions or a sufficient grid resolution are not finally answered, yet. On the other hand these type of models are more and more applied for short term air quality forecasting The present manuscript offers a sensitivity study conducted with the model CHIMERE, which was set up for the northwestern Iberian Peninsula, a region with complex topography and a long, structured coast line. Meteorological data from the WRF model was fed into CHIMERE, for the emissions the HTAP v2.2 inventory was used. Two different horizontal and two different vertical resolutions were tested

as well as two chemistry mechanisms (SPARC-07 A and Melchior mechanism). Model derived nitrogen dioxide, PM10, PM2.5 and ozone concentrations are discussed and compared to observational data from a regional air quality network based on statistical jmeasures. The comparisons were done for daily minimum and maximum values of those substances. The underlying investigation for the article is a straightforward sensitivity study with a pragmatic choice of varied parameters ($\sim$12 km and $\sim$4 km horizontal grid resolution and 10 and 20 vertical layers to 500 hPA). The comparisons between the model set-ups is done (for daily extreme values only) by using the bias, Pearson correlation and standard deviation ratio for the chemicals under investigation in relation to respective observed values. In addition, a mean absolute error is chosen to compare the runs to a chosen reference case (the computationally cheapest). The results of this quite "technical" study may be interesting for those intending to set up CHIMERE for purposes, for which computing resources are a limiting factor. The result section is dominated by describing point for point in words, what the figures show anyway. No deeper investigation and discussion of possible reasons for the discrepancies among model runs for the different the set-ups are offered. E.g. for the strong statement in the conclusions section that "CHIMERE's performance is very poor" it is in my opinion not sufficient to just speculate that the used emission inventory has deficiencies for the region: This definitely should be investigated (e.g. by consulting other inventories).

General comment: I doubt that the paper in its current form would be of great interest for the typical ACP readership. It fits not well into the journals scope. Neither the used procedures are sufficiently innovative nor the analysis of the results is deep enough to provide transferrable insights. The results, which might be interesting from a technical point of view, are addressing a specific region only, they could have wider implications for the modelling community in atmospheric sciences, if the analysis would look closer at the influence of the heterogeneous terrain and coastal flow effects on the findings. I leave a final consideration to the editor. In general, a more thorough discussion of reasons for the presented deviations between results from runs for the different model

set-ups and from the measurements is needed. Since the results have some value for air quality modelling, I would suggest to the authors considering a submission, though in a revised and extended form, to a journal, which is more devoted to technical analyses for modelling.

Some mayor points In addition to the remarks made above some further issues (shortcomings) of the manuscript need to be mentioned.

Emissions The backbone of air quality studies, especially when compared to observations, are suited emissions. The authors use HTAP v2.2 for the year 2010, while the study period are two summer months of 2018. A discussion of implications of this mismatch is missing. If the necessary observational data would be available, this technical study could have been performed for 2010 using appropriate meteorology. Or the 2010 emissions from HTAP could have been compared to more recent emission data and may be scaled (2010 compared to 2018). HTAP v2.2 emissions are provided on $0.1°$ x $0.1°$ grid, which does not directly fit the used resolution ($0.15°$ x $0.15°$ and $0.05°$ x $0.04°$). The regridding was done without downscaling. The authors do not explain what this mismatch means. But they should, since the resolution of the emissions may affect the results differently on the two grids, certainly a limitation of the study. HTAP v2.2 emissions are provided with a monthly time resolution. The authors do not inform the reader how they dealt with this coarse resolution when feeding the emissions into CHIMERE. Did they use time profiles on the emissions or did they feed in just the monthly means. A higher time resolution is needed, when comparing to daily maximum and minimum values of the observations (i.e. NO/NO2 and O3 relations are strongly dependent on the daily emission time profile). Any way, it is not adequate to state "CHIMERE's performance for NO2 is very poor" and point at the same time to possible deficiencies in the emissions (in that case it is not CHIMERE's performance). Page 17/ Line 16 (1 on that page) To improve their manuscript, the authors need to devote an entire section to these emission issues.

Meteorology The study area is characterized by structured terrain and a complex coast

line. The question appears, whether the meteorological model in use (here WRF) is resolving the local flow features sufficiently well (terrain effects and summer see breeze circulations)? The 12 km and 4 km runs might produce different results here, not unimportant, since quite a few of the observational stations are located near the cost or in hilly areas, where local flow fields might dominate the dispersion of emitted substances. Differences between modelled and observe concentration maxima/minima could partly be due to the quality of the meteorological simulations rather than entirely ascribed to CHIMERE. A discussion of the quality of the meteorological fields and reproduction of local features (best against observations) needs to be provided.

Data handling for results section The evaluation of the modelling results using observational data only considers maximum and minimum values. No information is provided how the values are taken from the respective series. There are several possibilities. Is the maximum taken from the observational time series and compared to the model result for the same time stamp? Or is the maximum taken from the observational time series and compared to the maximum of that day in the model series, which could occur at a different time (a considerable time shift might be possible). Or is the model output leading the selection? The same questions holds for the comparison of minimum values. It is interesting to learn whether maxima/minima are missed in general or whether there is a certain time shift. Although maximum values are important for air quality and health related studies, it would have been instructive to additionally analyse better time resolved concentrations to assess the models ability to reproduce daily cycles in different regions (i.e. O3) and the variability in the model and observational data. Both should be available with an hourly resolution. This could help also to discuss reasons for the deviations between the different set up. Show a few selected time series (for different quantities, different locations) of modelled versus measured concentrations (hourly resolution); may be more of that in the supplement. This would be very instructive. In an additional evaluation step the Mean Absolute Skill Score (using the reference run CS10) was provided separately for background, industry and traffic locations. In general no bad idea. It should be discussed, why for the background and

industry stations NO2 and O3 perform so differently for the different settings? Are they really that much decoupled? Are in case of the background stations O3 concentrations influenced predominantly by BVOCs from MEGAN? This needs a more thorough discussion. Quite a few of the stations used as bases for the statistical analysis are so called "traffic stations". These stations are often hotspots for some of the considered substances (NO2, PM10), because local traffic emissions are dominating (not resolved by HTAP). The authors should inform the readers about these traffic stations. Are some of them located within street canyons, which channel the flow and dispersion near the ground? This very local data is compared to model results obtained with a relatively coarse resolution (4 km and more horizontally). This seems not to be appropriate. It would be recommended to nit consider traffic station in the statistical evaluation (or do it separately to see the effect). Also for the other measurement stations, it would be useful to know, how they are located within the modelling grid. A station, which is located close to the grid boundary or grid corner, might be better represented by the neighbouring grid cell(s), dependent on local terrain effects. A study of this localisation effects for the 12 km and 4 km grid resolution would be helpful. Point measurements are compared to grid box means. The issue of the spatial representativeness of the stations needs to be addressed. Which of the background stations are in forested regions (BVOC emissions)?

Minor points

Since computational costs ares one of the parameters of interest/motivation of this study, it would be helpful for the reader to get quantitative information on the model runtime for the different experiments.

On page 8, line 18, CS10 is flagged as the computational cheapest experiment. In table 3, in which the experiments are ordered according to their computational costs, CM10 is the first mentioned and here apparently the cheapest. Clarify.
* * *
[Figure]

2019.

---

## Author Comment (AC1) · 10 Jan 2020

Response to Anonymous Referee #1

Referee comment: The manuscript of Brands et al. investigates the sensitivity of the CHIMERE model to different horizontal and vertical resolutions and to different chemical mechanisms. The focus is on a region of the northwestern Iberian Peninsula where the CHIMERE model has not been applied so far. The model results are evaluated against observations with a focus on minimum and maximum values. To my opinion the current version of this manuscript does not fit into the scope of ACP(D) as it is mainly a technical analysis of the model system and has no broader scope or gen-

eral implications for atmospheric science (see aims and scope of ACP; "The journal scope is focused on studies with general implications for atmospheric science rather than investigations that are primarily of local or technical interest"). To fit into the scope of ACP(D) a more general geoscientific conclusion/scope of the manuscript would be necessary. However, I think that the technical analysis of the model system is an interesting topic and therefore I propose to revise the manuscript largely (see detailed suggestions below) and hand the manuscript over to the partner journal Geoscientific Model Development (GMD). If the authors wish to publish their manuscript in ACP(D) the scope of the journal should be taken better into account and the manuscript needs major revisions.

Response: We would like to thank you very much for taking the time to review our manuscript and for your helpful and constructive criticism.

During the last months, we have undertaken an extensive review and have adressed all points raised by you not only by mere discussion but primarily by adding additional analyses and results. Most importantly, we have largely extended the number of model experiments (from 8 to 19), including 8 experiments run with the EMEP 2017 inventory, which is why it took a little bit longer to complete this review. To improve the interpretability and transferability of our results and to thrive away from the purely technical aspects, we have included additional figures, table content, text passages and conclusions that are of general interest for atmospheric science. While this should make our manuscript more relevant for the ACP readership, we would also agree on handing it over to GMD, if this is intended by the editor.

General comments

Referee comment: In p4 the authors write '[...]without using downscaling with traffic or population proxies[...]'. Accordingly, also for the finest nest with 0.05 x0.04 resolution, the authors applied emissions with a resolution of 0.1x0.1, right? If this is the case this is an important limitation of the study and needs to be clearly stated.

Response: This is a misinterpretation caused by us, arising from a too short description of the applied emission postprocessing in the first version of the manuscript. We surely did not run the 0.05° x 0.04° (lon x lat) CHIMERE domain with anthropogenic emissions on a 0.1° x 0.1° grid and have never tried to do so since the model would probably crash in this case. Rather, the raw emission from HTAP 2010 (and now also EMEP 2017) have been downscaled to the 0.05° x 0.04° grid with the help of different proxies, as described below. In the revised manuscript, an entire section is dedicated to the applied anthropogenic emission inventories and downscaling techniques (see Section 2.2):

Referee Comment: It is well known that the emission resolution influence the results largely. A too coarse resolution of the emissions can also deteriorate the model results (e.g. Markakis et al., 2015). Therefore, I propose to perform additional model runs with a downscaling of the emissions as this is a general feature of the CHIMERE model. These additional runs can then be used to quantify uncertainties due to missing downscaling of emissions.

Response: Already for the 8 experiments included in the first version of the manuscript, the raw HTAP 2010 emissions had been downscaled with land-use categories, i.e. with the basic downscaling option of the emiSURF program shipped with version 2017r4 of CHIMERE, but no further proxies had been considered. For the revised manuscript, 8 additional experiments have been performed with anthropogenic emissions from EMEP 2017, downscaled with population and traffic proxies, as well as with the location of large point sources. On top of the 16 HTAP and EMEP experiments, 3 additional experiments were run to answer specific questions. For one of these specific experiments, HTAP 2010 was downscaled with land-use categories and population density only, i.e. traffic density and large point sources were not used in this case in order to isolate the effect of the population proxy (see Table 3 in the revised manuscript for the detailed infos). The corresponding results can be found in Section 3.1.2, 3.2.2 and 3.3.

Referee comment: The authors find a poor performance for NO2 of the CHIMERE

model and link this poor performance to deficits of the emission inventory. To my opinion this argument needs a more detailed investigation. Several things should be discussed/considered:

Response: In the first version of the manuscript, the conclusions drawn on the model's performance for NO2 were too strong and have been downweighted in the revised manuscript. The large biases found for this species are due to the fact that the percentage bias is used in a region where the NO2 emissions at background sides are very low, part (see Figures 2 and 6). Consequently, a bias of only a few ug/m$^3$ in absolute terms translates into a large percentage value at these stations, which then largely increases the width of the boxplots (see e.g. Figure 8a). In the revised manuscript, this has been clarified in lines 376-378.

Referee comment: The emission inventory is for the year 2010. What was the emission changes in the last 8 years? What trend do ground-level measurements of NO2 show? Are there more up to date emission inventories available (for example from TNO,EDGAR etc.)? If so, why don't use them?

Response: To fix the problem of outdated emissions, the aforementioned 8 experiments based on EMEP 2017 were additionally conducted (see Section 2.3 and Table 3) and the respective results added to the revised manuscript (see Section 3). EMEP 2017 is the most up-to-date emission inventory provided by the European Union and the one year difference to our study period (summer 2018) is assumed to be negligible.

Referee comment: What is the performance of NO2 at the different stations types? How well does the model perform at the 'background' stations? How well at the "traffic" and "industry" stations?

Response: The performance of the four considered chemical species (including NO2) at background, traffic and industry sides was already shortly assessed in the first version of the manuscript (see Figure 6 therein). For the revised manuscript, this part of the study has been extended (see Figure 10 and Section 3.3 for the respective results).

Referee comment: Where are the "traffic" stations located? Does it make sense to evaluate a model with resolutions of 4 to 5 km and emissions at around 10 km resolution with measurements at the street scale? I guess it makes more sense to evaluate against the measurements at (urban) background stations or average all values of the 'traffic' stations of one particular city.

Response: This is a good point which not only applies to traffic stations but also to industrial sides or to any other sides affected by localized emissions. However, to our understanding, the downscaling techniques applied here have precisely been developed with the aim to improve CHIMERE's forecast skill on the local scale (Mailler et al. 2017) and, from this point of view, traffic or industrial stations should not be removed from the validation. To solve this dilemma, we do show the model's performance at traffic (and also industry) sides, which often are outliers if compared to the results at the remaining locations, but only in case they do not hamper the interpretation of the results. Consequently, outliers are shown in the overlay maps (Section 3.1.1 and 3.2.1) but ignored in the boxplots (Section 3.1.2 and 3.2.2) since their inclusion in the latter would blow up the scale of the x-axes.

Furthermore, we use outlier-resistant statistics such as the median instead of the arithmetic mean (see vertical orange line in the centre of the boxplots). The median model performance at traffic sides is shown in panels j, k and l of Figure 10. A description of the corresponding results can be found in Section 3.3. City-specific averages are an interesting alternative and will be considered in future studies.

Referee comment: How well does WRF reproduce the observed meteorology? Too efficient mixing of the boundary layer might cause problems in reproducing the measurements. Please provide at least a basic meteorological evaluation of the used meteorological data.

Response: For now more than a decade, the WRF configuration used to drive CHIMERE has been the backbone of the real-time forecasting system used by the

Galician governmental weather service MeteoGalicia. During the course of time, its performance has been steadily supervised by a team of operational forecasters as part of their day-to-day business and any large errors have been eliminated, if possible. In the revised manuscript, WRF's performance for a typical summertime heat day is illustrated in Supplementary Figure 1, which demonstrates that orographic and costal effects on the local weather are fairly well reproduced by the model (see also lines 95-99 of the revised manuscript).

Referee comment: The authors focus only on daily minimum and maximum values. I agree that especially the maximum values are very important with respect to air quality issues. However, to my opinion it would be very important to investigate also the general ability of the model to represent the hourly variability of the measurements. Therefore, I propose to further perform statistical analysis of the whole time series for each station and not only for minimum and maximum values.

Response: As suggested by you, a verification of the hourly time series has been included in Figure 10 of the revised manuscripts and the respective results are described in Section 3.3. We found that CHIMRE's performance for hourly data in many aspects is similar to the performance obtained for daily maxima.

Referee comment: Further, the analysis does not take into account that the model concentrations could be shifted geographically (e.g. minima and maxima are misplaced due to coarse resolution of the emissions). Therefore, I propose to provide additionally overlay plots (maybe only in the supplement) combining the geographical distribution of the modelled concentration and the measured concentrations as an example see Fig. 5 of Knote et al, 2011).

Response: To assess the possibility of geographical shifts in the modelled concentrations, and to present our results in a more illustrative manner, we have introduced a total of 60 overlay maps, distributed over 4 Figures (see Figure 2, 3, 6 and 7 and Section 3.1.1 and 3.2.1 for a description of the respective results). For the statistics

assessed in the present study (i.e. the temporal mean and standard deviation of the daily maximum and minimum time series), we have little indications of abrupt spatial shifts in the modelled values. Instead, the modelled values change rather smoothly from one grid-box to another and a shift of the modelled values by a few grid-boxes in any direction would lead to similar verification results.

Referee comment: I guess that the model runtime (and needed resources) of the different experiments differ heavily. It would therefore be very beneficial to confront the performance of the model with of the different configurations with the model runtime and to give recommendations about the trade-off between model runtime and model performance. This could have important implications for other people using CHIMERE. Further, if the authors (or others) plan to use the CHIMERE model for operational forecasts this trade-off would be very important information.

Response: Thanks for this valuable comment. In the last column of the revised Table 3, the runtime (in seconds) of the 8 experiments run with EMEP are listed for a typical summertime heat day. The respective values for the experiments conducted with HTAP are virtually identical, but cannot be stated since the exact values unfortunately have not been saved (see lines 192-194 of the revised manuscript). On the basis of this figure, the reader should be able to make a trade-off between model runtime and performance considering her/his own resources and requirements.

Referee comment: The description of the performed sensitivity studies (Sect. 2.2) is much too short. Readers familiar with the CHIMERE system might be able to follow the description of the authors; readers from outside the 'CHIMERE world' are lost. Please provide more details about the two different vertical grids (e.g. by a figure showing the different levels). Further, please describe the differences of the two chemical mechanisms in more detail. How do they differ? I know that Mailler et al., 2017 provides some details, but details which are very important for this study should be repeated in the manuscript. Further, Menut et al., 2013 already provide a short comparison of the MELCHIOR2 and SAPRC07 mechanisms. How do the findings from the authors

compare to the findings of Menut et al., 2013?

Response: As suggested by you, the description of the applied sensitivity tests has been extended and the orography and model layers for the two horizontal and vertical resolution set-ups have been added to Figure 1 of the revised manuscript. Also, the main conceptual differences between full Melchior and SAPRC have been stated and so are the main differences between the simulations run with reduced Melchior (or Melchior2) and SAPRC found in Menut et al. 2013. However, we here apply the full Melchior mechanism instead of the reduced one, meaning that a comparison with Menut et al. 2013 on equal terms is unfortunately not possible. After stating all this in lines 172-183 of the revised manuscript, we found larger average O3 maxima for SAPRC than for full Melchior, leading to larger positive biases for the former (see lines 312-313 and Figure 4e).

Specific comments

Referee comment: The line numbers in the manuscript seem to be wrong, at least on page 8. I here refer to the line numbers given in the manuscript

Response: In the revised manuscript, the line numbering has been fixed.

Referee comment: Abstract: Please provide information about the period consider for this investigation in the abstract (e.g. 20.7.2018-31.8.2018).

Response: In the revised manuscript, the study period is now stated right at the beginning of the abstract, as requested by you.

Referee comment: Section 2.1: Studies show that ozone on the Iberian Peninsula is heavily influenced by long range transport (e.g. Pay et al., 2019). Hence, boundary conditions are very important in this region. Therefore, please provide more information about the temporal update frequency of the boundary conditions in Sect. 2.1.

Response: In the revised manuscript, the influence of long range transport on the ozone conditions in the Iberian Peninsula is stated in lines 74-79 and the mentioned

study is cited. The boundary conditions data from C-IFS used in the present study are 3-hourly and this is stated in lines 115-116 of the revised manuscript.

Referee comment: Further, the authors mention that chemical boundary conditions stem from different systems (C-IFS, MACC). Further, the meteorological boundary conditions for WRF stem from GFS. Please provide short discussions about the influence of inconsistent chemical (and meteorological) boundary conditions.

Response: The fact that the boundary conditions are from different sources and that our WRF simulations are not forced with IFS but GFS data is something we have to live with since 1) C-IFS does not provide all chemical species necessary to run CHIMERE, which is why we also use MACC data and 2) we unfortunately do not have access to the meteorological forecasts from IFS, which is why we use GFS to force WRF instead. The impact these unavoidable inconsistencies have on CHIMERE's performance is expected to grow with increasing leadtime. Since the leadtime of the analysed time series is relatively short in our study (27 hours from initialization at the utmost), we assume it to be of minor importance here. This is pointed out in lines 124-126 of the revised manuscript.

Referee comment: Further, the authors mention that dust information from C-IFS needed to be scaled. What about the other components? Please provide short information about the quality of the chemical boundary data for the investigated period.

Response: In lines 115-126 of the revised manuscript, the chemical boundary data is now described with more detail. The scaling factor of 0.6 is only applied for desert dust. All other species taken from C-IFS are scaled by a factor of 1 (i.e. no scaling). The forecast skill of the C-IFS model is described in Flemming et al. 2015, which is also cited along these lines.

Referee comment: P6l10 The authors mention that for dust there is only a minor benefit when the model top height is increased. For ozone, however transport from the stratosphere is a very important feature which is missing in the applied set-up. Please

comment on this issue.

Response: By design of our experiments, stratospheric ozone intrusions are inherited from the C-IFS model in which the stratospheric O3 chemistry is parametrized, at least in the L60 model version used here (Flemming et al. 2015). Such intrusions, if present at all, were very weak during our study period (summer 2018) and the associated effects are thus assumed to be negligible.

Referee comment: P8l5 How did the authors sample the model data? The authors took the results at the lowest level in the corresponding grid box, right? But did the authors chose instantaneous model results or temporal (e.g. hourly averaged) model output?

Response: The standard hourly output from the CHIMERE simulations has been compared with hourly average observations. These hourly data were then used to compute the daily minima and maxima. In the revised manuscript, this is stated in lines 223-225.

Referee comment: P9l21 Especially in complex terrain the height of the lowest model layer and the height of the station might not fit together. Therefore, please provide a comparison of the station height and the height of the model at the lowest layer and check how large the differences are.

Response: We found little indications that the model errors at the higher stations, where height differences between the model and the real world are expected to be largest, significantly differ from those at the remaining stations of the same type (background). A more detailed assessment on this issue is interesting but out of the scope of the present paper. It will be undertaken in future studies.

Technical corrections

Referee comment: Figure 2: Please fix the legend ('hores')

Response: This issue has been fixed, thanks for careful reading.
* * *
**a) Maximum temperature (°C)**

**b) Minimum temperature (°C)**

**c) Maximum wind speed (km/h)**

**d) Average wind speed (km/h)**

**Fig. 1.** Suppl. Fig. 1) Daily observations vs. WRF forecasts for a August 5, 2018, (a) max. and
(b) min. temperature, (c) max. and (d) average wind speed

[Figure]

**NO2**

a)

b)

c)

d)

**O3**

e)

f)

g)

h)

**Fig. 2.** Suppl. Fig. 2) As Figure 4, but for background sides only

---

## Author Comment (AC2) · 10 Jan 2020

Response to Anonymous Referee #2

Referee comment: Quite a few different models are currently in use for chemistry transport modelling on the regional scale. Still many questions concerning the validity of the model results with respect to the necessary complexity of the chemistry mechanisms, the needed quality of underlying emissions or a sufficient grid resolution are not finally answered, yet. On the other hand these type of models are more and more applied for short term air quality forecasting. The present manuscript offers a sensitivity study conducted with the model CHIMERE, which was set up for the north-western Iberian

[Figure]

Peninsula, a region with complex topography and a long, structured coast line. Meteorological data from the WRF model was fed into CHIMERE, for the emissions the HTAP v2.2 inventory was used. Two different horizontal and two different vertical resolutions were tested as well as two chemistry mechanisms (SPARC-07 A and Melchior mechanism). Model derived nitrogen dioxide, PM10, PM2.5 and ozone concentrations are discussed and compared to observational data from a regional air quality network based on statistical measures. The comparisons were done for daily minimum and maximum values of those substances. The underlying investigation for the article is a straightforward sensitivity study with a pragmatic choice of varied parameters (âĹij12 km andâĹij4 km horizontal grid resolution and 10 and 20 vertical layers to 500 hPa). The comparisons between the model set-ups is done (for daily extreme values only) by using the bias, Pearson correlation and standard deviation ratio for the chemicals under investigation in relation to respective observed values. In addition, a mean absolute error is chosen to compare the runs to a chosen reference case (the computationally cheapest). The results of this quite "technical" study may be interesting for those intending to set up CHIMERE for purposes, for which computing resources are a limiting factor. The result section is dominated by describing point for point in words, what the figures show anyway. No deeper investigation and discussion of possible reasons for the discrepancies among model runs for the different the set-ups are offered. E.g. for the strong statement in the conclusions section that "CHIMERE‒s performance is very poor" it is in my opinion not sufficient to just speculate that the used emission inventory has deficiencies for the region: This definitely should be investigated (e.g. by consulting other inventories).

Response: We would like to thank you very much for taking the time to review our manuscript and for sharing your thoughts with us. During the last months, we have made large efforts to take into account your valuable critics and suggestions and this is why it took a little more time to complete this review. Most importantly, we ran all model experiments once again with another emission configuration (EMEP 2017 downscaled with several proxies) and conducted even more sensitivity experiments

to assess the specific roles of 1) the population proxy used for downscaling raw anthropogenic emissions, 2) the resolution of the meteorological input data from WRF and 3) biogenic emissions. We added new figure and table material, rewrote nearly the entire manuscript and now provide a more profound interpretation of our findings. Also, after an reassessment of the corresponding results, the text passage stating that "CHIMERE's performance is very poor..." was removed since it no longer holds.

General comments

Referee comment: I doubt that the paper in its current form would be of great interest for the typical ACP readership. It fits not well into the journals scope. Neither the used procedures are sufficiently innovative nor the analysis of the results is deep enough to provide transferable insights. The results, which might be interesting from a technical point of view, are addressing a specific region only, they could have wider implications for the modelling community in atmospheric sciences, if the analysis would look closer at the influence of the heterogeneous terrain and coastal flow effects on the findings. I leave a final consideration to the editor. In general, a more thorough discussion of reasons for the presented deviations between results from runs for the different model setups and from the measurements is needed. Since the results have some value for air quality modelling, I would suggest to the authors considering a submission, though in a revised and extended form, to a journal, which is more devoted to technical analyses for modelling.

Response: The revised manuscript has been nearly complete rewritten, adding new figure and table material as well as an interpretation of our findings whenever we think this is justified from our experimental analyses (e.g. in lines 265-271, 376-378, 481-492 and 516-523) However, despite our efforts to thrive away from a purely technical paper in order to take into account your suggestions, the principal aim of our study is not to go deep with specific meteorological phenomena or the with differences between specific experiment pairs, but to estimate the role of as many parameters as possible on CHIMERE's average performance when compared to observations in order to make

recommendations on the best model configuration. This is crucial for setting up reliable model simulations, be it for real-time forecasts or in analysis mode, i.e. whenever CHIMERE is assumed to realistically reproduce the real world. To our knowledge, such an endeavour has not been undertaken yet for this type of study region (mid-latitudes, coastal, complex orography, relatively low pollution) albeit regional air quality prediction schemes are demanded there by politics, scientists and the general public.

Due the "broad" approximation or our study, an in-depth analysis for each of the many differences found between the experiments cannot be provided in one single study. However, thanks to the large extension of applied numerical experiments we are now in the position to provide an extended list of conclusions, some of which are not only of purely technical but rather general interest from our point of view (see Section 4 of the revised manuscript).

In summary, we are confident that our study is of interest for ACP and also would like to mention that this journal has published similar studies in the past (e.g. Bessagnet et al. 2016, see reference list of the revised manuscript). We would of course also welcome handing the manuscript over to GMD, should the editor decide so.

Some mayor points

Referee comment: In addition to the remarks made above some further issues (short-comings) of the manuscript need to be mentioned.

Emissions: The backbone of air quality studies, especially when compared to obser-vations, are suited emissions. The authors use HTAP v2.2 for the year 2010, while the study period are two summer months of 2018. A discussion of implications of this mis-match is missing. If the necessary observational data would be available, this technical study could have been performed for 2010 using appropriate meteorology. Or the 2010 emissions from HTAP could have been compared to more recent emission data and may be scaled (2010 compared to 2018).

Response: To solve this problem, a total of 8 new experiments were run with the anthropogenic emission inventory EMEP 2017, which is the most up-to-date inventory provided by the European Union (see Section 2.2). It is then straightforward to assume that the effects arising from the one year difference to our the study period are negligible.

Referee comment: HTAP v2.2 emissions are provided on 0.1â٧x 0.1â٧grid, which does not directly fit the used resolution (0.15â٧x 0.15â٧and 0.05â٧x 0.04â٧). The regridding was done without downscaling. The authors do not explain what this mismatch means. But they should, since the resolution of the emissions may affect the results differently on the two grids, certainly a limitation of the study.

Response: We are afraid this is a misinterpretation arising from a too short description of the emission postprocessing in the first version of the manuscript. In fact, the 8 experiments shown in the first version already had been downscaled to the 0.05° x 0.04 grid using land-use categories, which is the basic option of the emiSURF emission postprocessor shipped with CHIMERE. We never tried to run CHIMERE with emissions on another grid and think the model would probably crash when trying to do so.

Referee comment: HTAPv2.2 emissions are provided with a monthly time resolution. The authors do not inform the reader how they dealt with this coarse resolution when feeding the emissions into CHIMERE. Did they use time profiles on the emissions or did they feed in just the monthly means? A higher time resolution is needed, when comparing to daily maximum and minimum values of the observations (i.e. NO/NO2 and O3 relations are strongly dependent on the daily emission time profile).

Response: Similarly, already for the experiments in the first version of the manuscript, monthly HTAPv2.2 emissions had been disaggregated to hourly timescale using the standard information in the CHIMERE pre-processors (Mailler et al. 2017). In the revised manuscript, this is stated in lines 151-154. Referee comment: Any way, it is not adequate to state "CHIMERE's performance for NO2 is very poor" and point

at the same time to possible deficiencies in the emissions (in that case it is not CHIMERE′s performance). Page 17/Line 16 (1 on that page). To improve their manuscript, the authors need to devote an entire section to these emission issues.

Response: You are absolutely right, this conclusion has been removed from the manuscript. Most of the large percentage bias values for NO2 are located at background stations where the observed mean concentrations for this species are generally very low, meaning that an absolute bias of only a few ug/m$^3$ (which is obviously not important) translates into a large percentage value. This has to be considered when interpreting the results and is stated in lines 376-378 of the revised manuscript.

Referee comment: Meteorology The study area is characterized by structured terrain and a complex coastline. The question appears whether the meteorological model in use (here WRF) is resolving the local flow features sufficiently well (terrain effects and summer sea breeze circulations)? The 12 km and 4 km runs might produce different results here, not unimportant, since quite a few of the observational stations are located near the coast or in hilly areas, where local flow fields might dominate the dispersion of emitted substances. Differences between modelled and observed concentration maxima/minima could partly be due to the quality of the meteorological simulations rather than entirely ascribed to CHIMERE. A discussion of the quality of the meteorological fields and reproduction of local features (best against observations) needs to be provided.

Response: The WRF configuration used to force the CHIMERE experiments has been in operational use at the Galician Meteorological Service (MeteoGalicia) for now more than a decade. Its performance is supervised by a team of operational forecasters as part of their day-to-day business and any large model deficiencies, if possible, have been corrected in the course of time. In Supplementary Figure 1, the performance of the model is illustrated for a typical summertime heat day for the high-resolution domain (4 km), showing that local wind and temperature features are reproduced fairly well by the model. In the revised manuscript, this is pointed out in lines 95-99.

Furthermore, to assess the influence the "meteorological resolution" has on CHIMERE's performance, an additional experiment was designed in which the fine CHIMERE domain (0.05° x 0.04°) was not run with the 4 km WRF simulation (as in all other experiments) but with the coarse 12 km simulation (see Table 3). If the regional orography, the structure of the coastline or the land-sea contrast was key for CHIMERE's performance, then this "coarse meteorology" experiment should produce considerably worse results than the reference experiment run with fine-scale meteorological input (compare FM20H-C with FM20H in the boxplots). However, our results indicate that the performance differences between these two experiments are generally smaller than expected and that other factors are more influential on model performance (see lines 304-307, 336-337, 401-402 and 516-523).

Referee comment: Data handling for results section. The evaluation of the modelling results using observational data only considers maximum and minimum values. No information is provided how the values are taken from the respective series. There are several possibilities. Is the maximum taken from the observational time series and compared to the model result for the same time stamp? Or is the maximum taken from the observational time series and compared to the maximum of that day in the model series, which could occur at a different time (a considerable time shift might be possible). Or is the model output leading the selection? The same questions holds for the comparison of minimum values. It is interesting to learn whether maxima/minima are missed in general or whether there is a certain time shift.

Response: The modelled daily extreme values are calculated on the hourly model data for a given day and the observed daily extreme values are calculated on the observed hourly data for this day, i.e. the calculation for the model and observations is independent from each other meaning that the extremes can occur at different times of day. In the revised manuscript, this is stated in lines 223-225. Albeit a detailed assessment of the modelled vs. observed daily cycle is out of the scope of the present study, in the revised manuscript we now as well include the verification results of the

hourly data (see Section 3.3 and Figure 10).

Referee comment: Although maximum values are important for air quality and health related studies, it would have been instructive to additionally analyse better time resolved concentrations to assess the models ability to reproduce daily cycles in different regions (i.e. O3) and the variability in the model and observational data. Both should be available with an hourly resolution. This could help also to discuss reasons for the deviations between the different set ups. Show a few selected time series (for different quantities, different locations) of modelled versus measured concentrations (hourly resolution); may be more of that in the supplement. This would be very instructive.

Response: In Section 3.3 and Figure 10 of the revised manuscript, we have included the verification results for hourly data which provides insight on the average performance of the distinct model experiments on sub daily timescale. In many aspects, the results for the hourly data are similar to those obtained for the maxima. As argued at the start of this response letter, an in-depth analysis of the daily cycles is one of the efforts to be undertaken in future studies since the manuscript size is already at its upper limit.

Referee comment: In an additional evaluation step the Mean Absolute Skill Score (using the reference run CS10) was provided separately for background, industry and traffic locations. In general no bad idea. It should be discussed, why for the background and industry stations NO2 and O3 perform so differently for the different settings? Are they really that much decoupled? Are in case of the background stations O3 concentrations influenced predominantly by BVOCs from MEGAN? This needs a more thorough discussion.

Response: For the revised manuscript, we have performed an additional model experiment in which the biogenic emissions (comprising biogenic VOCs and NO) were intentionally turned off. We then specifically looked at the model performance for the background O3 and NO2 maxima (see Supplementary Figure 2). At these sides, both

the O3 and NO2 maxima are reduced when biogenic emissions are intentionally turned off, meaning that the concentration of both species is affected by this emission type. In the revised manuscript, this is pointed out in lines 477-493.

Referee comment: Quite a few of the stations used as bases for the statistical analysis are so called "traffic stations". These stations are often hotspots for some of the considered substances (NO2, PM10), because local traffic emissions are dominating (not resolved by HTAP). The authors should inform the readers about these traffic stations. Are some of them located within street canyons, which channel the flow and dispersion near the ground?

Response: The traffic stations from the Galician air quality monitoring network are not located in street canyons but rather in relatively open terrain, as required by the Spanish legislation. We are well aware that our CHIMERE experiments run at a resolution of 0.05° x 0.04° do not resolve concentration differences on the street scale but it is nevertheless interesting to know what the model suggests for these stations, particularly taking into account that the downscaling procedure applied to the EMEP emissions uses road traffic density as a proxy to re-allocate the raw emissions on the sub grid scale. Hence, we do not rule out the corresponding results unless they produce such large outliers that they hamper the visualization, in which case these outliers are simply not shown (as in the boxplots). We also use outlier-resistant statistics such as the median instead of the arithmetic mean (in the boxplots and Figure 10).

Referee comment: This very local data is compared to model results obtained with a relatively coarse resolution (4 km and more horizontally). This seems not to be appropriate. It would be recommended to nit consider traffic station in the statistical evaluation (or do it separately to see the effect).

Response: A separate evaluation of model performance for each station type (background, traffic, industry) is provided in Section 3.3 and Figure 10 of the manuscript. The corresponding text passages have been largely extended to discuss the corresponding

differences.

Referee comment: Also for the other measurement stations, it would be useful to know, how they are located within the modelling grid. A station located close to the grid boundary or grid corner might be better represented by the neighbouring grid cell(s), dependent on local terrain effects. A study of this localisation effects for the 12 km and 4 km grid resolution would be helpful.

Response: In the revised manuscript, a total of 60 maps are provided (in 4 Figures) in which the observed concentrations are plotted on top of the modelled ones. These "overlay maps" show that the modelled concentrations change relatively smoothly in space even on the 4 km grid. Hence, a hypothetical shift of the mesh by a few grid-boxes in any direction would return similar results.

Referee comment: Point measurements are compared to grid box means. The issue of the spatial representativeness of the stations needs to be addressed. Which of the background stations are in forested regions (BVOC emissions)?

Response: The fact that areal mean values from a model are compared to point measures is a common issue in the evaluation of these models and can't be overcome unless the modelled values are statistically downscaled to match the observations , in which case the bias could be probably further reduced. In our study area, almost all background stations are in forested area and thus influenced by BVOC emissions. In the revised manuscript, this is stated in lines 68-69 and line 103.

Minor points

Referee comment: Since computational costs are one of the parameters of interest/motivation of this study, it would be helpful for the reader to get quantitative information on the model runtime for the different experiments.

Response: We have included the runtimes for the 8 experiments run with EMEP in the last column of revised Table 3. The runtimes for the respective experiments run with

HTAP are nearly identical, but were unfortunately not saved. This is clarified in lines 191-194 of the revised manuscript.

Referee comment: On page 8, line 18, CS10 is flagged as the computational cheapest experiment. In table 3, in which the experiments are ordered according to their computational costs, CM10 is the first mentioned and here apparently the cheapest. Clarify.

Response: Thanks for careful reading, this error has been corrected. CS10 was the computationally cheapest experiments. In the revised manuscript, the ordering according to the computational costs only applies within the same emission configuration. Within the EMEP group (ending on "E"), CS10E is the cheapest experiment. Within the HTAP group (ending on "H") CS10H is the cheapest one, and so on.

**a) Maximum temperature (°C)**

**b) Minimum temperature (°C)**

**c) Maximum wind speed (km/h)**

**d) Average wind speed (km/h)**

**Fig. 1.** Suppl. Fig. 1) Daily observations vs. WRF forecasts for a August 5, 2018, (a) max. and(b) min. temperature, (c) max. and (d) average wind speed

[Figure]

**Fig. 2.** Suppl. Fig. 2) As Figure 4, but for background sides only